

# Effects of arginine vasopressin on the urine proteome in rats

Manxia An[1], Yanying Ni[1], Xundou Li[1] and Youhe Gao[1,2]

[1] Department of Pathophysiology, Institute of Basic Medical Sciences Chinese Academy of Medical Sciences, School of Basic Medicine Peking Union Medical College, Beijing, China
[2] Department of Biochemistry and Molecular Biology, Beijing Normal University, Gene Engineering and Biotechnology Beijing Key Laboratory, Beijing, China

## ABSTRACT

Biomarkers are the measurable changes associated with a physiological or pathophysiological process. The content of urine frequently changes because it is not controlled by homeostatic mechanisms, and these alterations can be a source of biomarkers. However, urine is affected by many factors. In this study, vasoconstrictor and antidiuretic arginine vasopressin (AVP) were infused into rats using an osmotic pump. The rats' urinary proteome after one week of infusion was analyzed by label-free LC-MS/MS. A total of 408 proteins were identified; among these proteins, eight and 10 proteins had significantly altered expression in the low and high dose groups, respectively, compared with the control group using the one-way ANOVA analysis followed by post hoc analysis with the least significant difference (LSD) test or Dunnett's T3 test. Three differential proteins were described in prior studies as related to AVP physiological processes, and nine differential proteins are known disease biomarkers. Sixteen of the 17 differential proteins have human orthologs. These results suggest that we should consider the effects of AVP on urinary proteins in future urinary disease biomarker researches. The study data provide clues regarding underlying mechanisms associated with AVP for future physiological researches on AVP. This study provide a sensitive changes associated with AVP. However, the limitation of this result is that the candidate biomarkers should be further verified and filtered. Large clinical samples must be examined to verify the differential proteins identified in this study before these proteins are used as biomarkers for pathological AVP increased diseases, such as syndrome of inappropriate antidiuretic hormone secretion (SIADH).

# INTRODUCTION

Biomarkers are the measurable changes associated with a physiological or pathophysiological process (*Gao, 2013*). Blood is under the homeostatic control of the body. Renal filtration, reabsorption of solutes, and excretion of water are examples of homoeostatic control of blood which in turn affect the composition of urine. Without homeostasis, urine contains many changes. Therefore, urine should be a better source of biomarkers. Additionally, urine can be obtained easily and noninvasively. It has been used in studies to detect disease biomarkers, such as kidney disease (*Pejcic, Stojnev & Stefanovic, 2010*), cardiovascular disease (*Delles, Diez & Dominiczak, 2011*) and liver disease (*Trovato et al., 2015*). However,

Corresponding author
Youhe Gao, gaoyouhe@bnu.edu.cn,
gaoyouhe@pumc.edu.cn

many non-disease factors, including age, gender, lifestyle and medications, can affect changes in urine (*Wu & Gao, 2015*). These factors can interfere or mask changes caused by the disease itself, increasing the difficulty of identifying reliable and disease-specific biomarkers. In clinical studies, age, gender and other factors can be balanced to a certain degree by the experimental design. The influence of medications is difficult to assess and balance between the experimental and control groups because only patients in the disease group usually receive medication treatment. It is possible that changes caused by medication are incorrectly considered to be disease biomarkers. Therefore, the effect of medications should be separately studied to facilitate identifying disease-specific biomarkers.

AVP, which is also called antidiuretic hormone, is a polypeptide hormone consisting of nine amino acids (*Kounin & Bashir, 2000*). It is synthesized and secreted in the supraoptic and paraventricular nuclei of the hypothalamus and is released into the blood to participate in physical activities because of certain physiological stimuli. Its two most important physiological functions are as a vasoconstrictor and an antidiuretic (*Kounin & Bashir, 2000*). AVP is also involved in a variety of physiological processes, including physiological stress (*Coverdill et al., 2012*; *Zelena et al., 2009*), memory (*Nabe et al., 2007*), thermoregulation (*Bicego-Nahas et al., 2000*) and pain regulation (*Madrazo et al., 1987*). In clinical applications, AVP is used for the treatment of many diseases, such as diabetes insipidus, hepatorenal syndrome, portal venous hypertension, bleeding disorders, septic shock and cardiopulmonary resuscitation (*Treschan & Peters, 2006*).

SIADH has common clinical symptoms and can be complicated by diseases characterized by abnormal increases in AVP secretion (*Frouget, 2012*). Excessive AVP secretion can promote the opening of water channels in the renal collecting duct and distal convoluted tubule (DCT), thereby increasing water reabsorption, reducing urine volume, increasing urine osmolality and decreasing serum sodium levels. Continuous AVP infusion can be used to establish a rat SIADH model for identification of SIADH urinary markers (*Verbalis, 1984*).

Studying the effects of AVP on urine can achieve the following goals: (1) providing some clues for understanding its physiological functions; (2) establishing a reference for urinary biomarker research when AVP is used as a medication in studies; and (3) identifying potential biomarkers for pathological increases in AVP in diseases such as SIADH.

In this study, AVP (10 ng/h and 50 ng/h) was infused into rats with an osmotic pump. The urine from the model rats and the controls was analyzed by label-free LC-MS/MS one week after AVP infusion. Using rat models can minimize the influence of other factors by strictly controlling the experimental conditions so that the AVP is the single influential factor. Another advantage of using rat models is that more reliable candidate biomarkers can be obtained from a small number of samples (*Gao, 2014*).

## MATERIALS AND METHODS

### Animal experiments

Male Sprague-Dawley rats (160–180 g) were purchased from the Institute of Laboratory Animal Science, Chinese Academy of Medical Science (Beijing, China). The rats were fed a standard laboratory diet and had free access to water. They were housed in standard

temperature ($22 \pm 1\,°C$) and humidity (65%–70%) conditions. The animal experiments were approved by the Institute of Basic Medical Sciences Animal Ethics Committee, Peking Union Medical College (Animal Welfare Assurance Number: ACUC-A02-2013-015).

The rats in control group were infused with normal saline ($n = 7$), the rats in low dose group were infused with low dose AVP (10 ng/h, $n = 5$), and the rats in high dose group were infused with high dose AVP (50 ng/h, $n = 6$). The normal saline and two different concentrations of AVP (Sigma-Aldrich, St. Louis, MO, USA, dissolved in normal saline) were continuously infused to rats by osmotic pump respectively. The surgical procedure was performed as follows. The rats were fasted with free access to water for 12 h before the experiment. The rats were anesthetized by 2% pentobarbital sodium (40 mg/kg) injection. The dorsal skin of the rats' necks was disinfected with povidone-iodine and cut laterally around a 1 cm incision. The skin and subcutaneous tissue within the incision were separated with blunt tweezers. An ALZET (Cupertino, CA, USA) osmotic pump (model 2002; reservoir volume of 200 µL, and flow rate of 0.5 µL/h) containing 200 µL of 20 ng/µL AVP, 100 ng/µL AVP, or normal saline was implanted into the incision. The incisions were sutured and disinfected with povidone-iodine after the osmotic pump was confirmed as occupying a slack space in the subcutaneous layer.

Rats of all groups were housed in common feeding cages with four to six rats in one cage. Urine was collected in metabolic cages one week after the AVP infusion, and the volume of urine was recorded. The rats were individually placed in a metabolic cage only during the urine collection. Rats were fasted and allowed free access to water during urine collection. All rats were subjected to the same conditions and most likely experienced similar stress levels during urine collection. No obvious injuries to the rats were observed during the urine collection in the metabolic cages. The body weights of the rats after the AVP infusion and one week after the AVP infusion were recorded. Physiological indicators of urine, including osmolality, total protein and creatinine, were measured at the Beijing Union Medical College Hospital. Due to the limited throughput of our mass spectrometry capacity in the laboratory, three rats were selected randomly from each group for urinary proteome profiling by LC-MS/MS.

## Extraction of urinary protein

Urine was centrifuged at 2,000 g for 30 min immediately after collection to remove the cell debris. The supernatant was centrifuged at 12,000 g for 30 min. Three volumes of acetone were added to the supernatant after removing the pellets, and precipitation was allowed to occur overnight at $-20\,°C$ followed by centrifugation at 12,000 g for 30 min. The pellets were resuspended in lysis buffer (8 M urea, 2 M thiourea, 50 mM Tris, and 25 mM dithiothreitol (DTT)) for 2 h at $4\,°C$. The solution was centrifuged at 12,000 g for 30 min and the supernatant was collected. The protein concentrations were determined using the Bradford method.

## SDS-PAGE analysis

Thirty micrograms of protein from each sample were mixed with sample buffer and incubated at $96\,°C$ for 10 min. The protein samples were then loaded onto 12% SDS-PAGE. The gel was stained using Coomassie Brilliant Blue.

## Urine sample preparation and LC-MS/MS

Urinary proteins were digested with trypsin using the filter-aided sample preparation method (*Wisniewski et al., 2009*). One hundred micrograms of protein were deposited onto a 10 kD filter membrane (Pall, Port Washington, NY, USA). Then, urea buffer (UA; 8 M urea and 0.1 M Tris-HCl, pH 8.6) and ammonium bicarbonate (25 mM) were added to the membrane and centrifuged at 14,000 g for 40 min at 18 °C to wash the samples. The samples were reduced by incubation with 20 mM dithiothreitol at 37 °C for 1 h and alkylated by 50 mM in iodoacetamide (IAA) in the dark for 40 min. UA and ammonium bicarbonate were added and centrifuged to remove the remaining DTT and IAA. Mass spec grade trypsin (Trypsin Gold, Mass Spec Grade; Promega, Fitchburg, WI, USA) was added to the filter membrane at an enzyme-to-protein ratio of 1:50 and incubated at 37 °C for 13 h. The digested peptides were obtained by centrifugation, desalted with Oasis HLB cartridges (Waters, Milford, MA, USA) and dried by vacuum evaporation.

One microgram of peptides was loaded onto a reversed-phase microcapillary column by EASY-nLC 1200 UHPLC system and eluted with a gradient of 5–28% mobile phase B (0.1% formic acid and 99.9% acetonitrile; flow rate of 0.3 mL/min) for 60 min. The eluted peptides were analyzed by Thermo Orbitrap Fusion Lumos MS (Thermo Fisher Scientific, Bremen, Germany). Each peptide sample was analyzed three times as technical replicates.

## MS data analysis

MS data were retrieved using the mascot software (version 2.5.1, Matrix Science, London, UK) and searched against the Swissprot_2014_07 database (taxonomy: Rattus, containing 7,787 sequences). Trypsin was selected as the digestion enzyme, and two missed trypsin cleavage sites were allowed. Carbamidomethylation of cysteines was selected as a fixed modification. The fragment mass tolerance was set to 0.6 Da, and the parent mass tolerance was set to 10 ppm. Mascot search results were screened and integrated using scaffold software (version 4.4.8; Proteome Software Inc., Portland, Oregon, USA). The peptide identification and protein identification false discovery rates were set to less than 1%, with each protein containing at least 2 identified peptides. Differential proteins between the control and AVP infusion groups were identified by quantitatively analyzing spectral counts. The Ensembl Compare database was searched to identify orthologous human proteins for these differential urinary proteins as reported (*Vilella et al., 2009*), and the human orthologs were then compared with the human core urinary proteome (*Nagaraj & Mann, 2011*).

## Hierarchical clustering of quantitative data

Hierarchical cluster analysis was used to understand the overall changes in proteins between groups and assess the parallelism and variations among technical replicates. An average-linkage hierarchical clustering of the top 280 proteins (after excluding proteins with low spectral count) was performed and visualized by R's gplot v3.01 package to create heat maps via the heatmap.2, the default hierarchical clustering method in hclust is "complete method".

### Biological function analysis of differential proteins

Biological functions of differential proteins, including molecular functions, biological processes and cellular components, were analyzed using the PANTHER classification system (http://www.pantherdb.org/).

### Statistical analysis

The $P$ value was analyzed by the Statistical Package for Social Studies (SPSS) 22.0. The differences in physiological indicators and urinary proteins were assessed using the SPSS software by one-way ANOVA followed by post hoc analysis with the least significant difference (LSD) test or Dunnett's T3 test. The difference between groups was considered significant when the $P$ value was equal to or less than 0.05.

## RESULTS

### Weight gain and urine protein-to-creatinine ratio

There were no significant difference of weight gains in the normal, low-dose, and high-dose groups by one week after AVP infusion (Fig. 1A). The urine protein-to-creatinine ratio was slightly higher in the low-dose group and the high-dose group than in the control group, although these differences were not statistically significant (Fig. 1B).

### Urine volume and urinary osmolality

The 24 h urine volume of rats in the AVP infusion group was significantly less than that of those in the control group, and the urine volume of the rats in the high-dose group was significantly less than that of those in the low-dose group (Fig. 1C). The urinary osmolality of rats in the AVP infusion group was significantly higher than that of those in the control group. Additionally, urinary osmolality of rats was higher in the high-dose group than in the low-dose group, although this difference was not statistically significant (Fig. 1D).

### SDS-PAGE analysis

Thirty microgram of urinary proteins were analyzed in SDS-PAGE to observe the protein distribution. The SDS-PAGE gel showed that there was no significant degradation of these samples (Fig. 2). The gel bands were not excised for mass spectrometric analysis. A different aliquot (100 micrograms) of urinary proteins were analyzed by mass spectrometry. No bands with consistent differences between the control and AVP infusion groups were observed in the SDS-PAGE analysis (Fig. 2).

### Proteomics analysis

Urine specimens of three rats from the control, low-dose and high-dose groups were analyzed by LC-MS/MS. Each sample was analyzed three times as technical replicates. Differential proteins were identified by semi-quantitative analyses of spectral counts (*Liu, Sadygov & Yates, 2004*; *Old et al., 2005*; *Schmidt et al., 2014*). The spectral count for each protein was calculated based on the mean spectral count of the proteins from three replicates.

A total of 408 proteins were identified in the three groups. The identified proteins are shown in Table S1. Forty-nine proteins significantly changed between control group

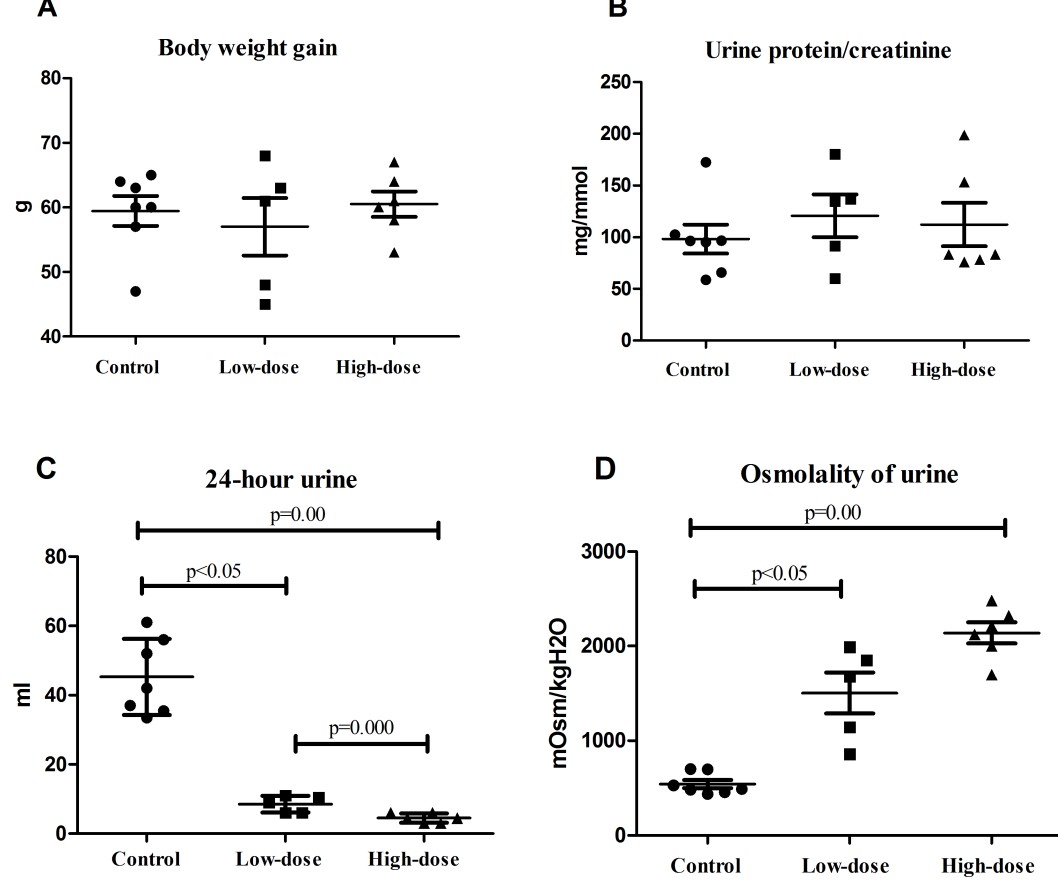

**Figure 1** **Physiological indicators of rats in control group ($n = 7$), low dose AVP group ($n = 5$) and high dose AVP group ($n = 6$).** (A) Weight gain of rats in three groups; (B) urine protein-to-creatinine ratio of rats in three groups; (C) 24 h urine volume of rats in three groups; (D) urinary osmolality of rats in three groups.

and AVP infusion groups ($P$ by ANOVA $\leq 0.05$). Compared to the control group, 21 differential proteins in the low-dose group and 37 in high-dose group were significantly different. Nine proteins were significantly changed in both groups.

To identify the most significantly differential protein, the following more stringent criteria were chosen: (1) $P$ by ANOVA $\leq 0.05$, (2) fold change $\geq 1.5$, and (3) spectral count for each sample $\geq 4$ in at least one group. Seventeen differential proteins were identified between control group and AVP infusion groups by these criteria. Eight differential proteins in the low-dose group and 10 in the high-dose group were identified relative to the control group (Tables 1 and 2). One protein was significantly changed in both groups.

### Hierarchical clustering of quantitative data
As indicated in Fig. 3, the high-dose group was easily distinguishable from the low-dose group and the control group. The three technical replicates for each sample can also be readily identified on the heatmap.

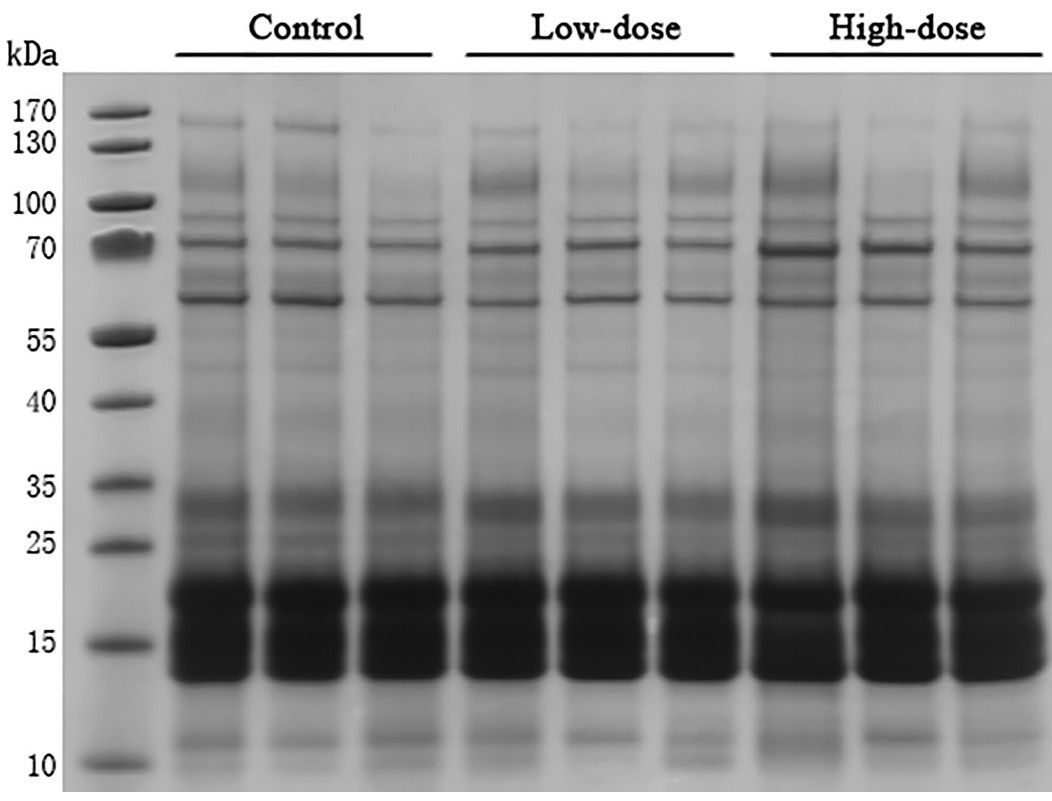

**Figure 2** SDS-PAGE of urinary proteins in control group, low dose AVP group and high dose AVP group. M, marker.

## Biological function analysis of differential proteins

Biological functions of 49 differential proteins that significantly changed between control group and AVP infusion groups were analyzed. The major molecular functions of the differential proteins include catalytic activity, binding and receptor activity. The differential proteins' main biological processes include cellular processes, biological regulation and developmental processes. The differential proteins' main cellular components include cell parts, organelles and extracellular regions (Fig. 4).

## Differential proteins reported to be associated with AVP

Three differential proteins have been reported to be related to the functions of AVP. Increased expression of osteopontin in rat aortic adventitial fibroblasts can be induced by urotensin II (*Zhang et al., 2011*), which has vasoconstrictive functions similar to those of vasopressin. In this study, osteopontin in urine was 6-fold higher in the high-dose group than in the control group.

Calbindin, as $Ca^{2+}$-buffer protein, can buffer $Ca^{2+}$ concentrations within cells and thereby prevent cell injuries caused by high intracellular $Ca^{2+}$ (*Schwaller, 2009*). Calbindin localized to the DCT and the connecting tubule can regulate calcium transport and reabsorption (*Hsin et al., 2006*; *Lambers et al., 2006*).

**Table 1** **The significant differential proteins between control group and low dose group.** Rat 1, 2, 3, control group; rat 4, 5, 6, low-dose group; fold change is low dose-to-control group ratio.

| Protein name | Accession number | Fold change | P value | Spectral counts | | | | | | Candidate biomarker |
|---|---|---|---|---|---|---|---|---|---|---|
| | | | | Rat 1 | Rat 2 | Rat 3 | Rat 4 | Rat 5 | Rat 6 | |
| Haptoglobin | HPT_RAT | 2.33 | 0.022 | 6 | 3 | 3 | 10 | 8 | 10 | Yes |
| Prolactin-inducible protein homolog | PIP_RAT | 1.88 | 0.009 | 2 | 3 | 3 | 5 | 6 | 4 | No |
| Pro-cathepsin H | CATH_RAT | 1.88 | 0.041 | 2 | 3 | 3 | 5 | 5 | 5 | Yes |
| Junctional adhesion molecule A | JAM1_RAT | 0.66 | 0.005 | 8 | 9 | 7 | 5 | 6 | 5 | No |
| Regenerating islet-derived protein 3-gamma | REG3G_RAT | 0.60 | 0.011 | 18 | 16 | 14 | 8 | 13 | 8 | Yes |
| Glutamate–cysteine ligase catalytic subunit | GSH1_RAT | 0.60 | 0.020 | 11 | 11 | 8 | 4 | 8 | 6 | No |
| Attractin | ATRN_RAT | 0.56 | 0.041 | 5 | 7 | 4 | 2 | 4 | 3 | No |
| Malate dehydrogenase, cytoplasmic | MDHC_RAT | 0.46 | 0.005 | 4 | 6 | 5 | 3 | 2 | 2 | No |

An et al. (2017), *PeerJ*, DOI 10.7717/peerj.3350

**Table 2 The significant differential proteins between control group and high dose group.** Rat 1, 2, 3, control group; rat 7, 8, 9, high-dose group; fold change is high dose-to-control group ratio.

| Protein name | Accession number | Fold change | P value | Spectral counts | | | | | | Candidate biomarkers |
|---|---|---|---|---|---|---|---|---|---|---|
| | | | | Rat 1 | Rat 2 | Rat 3 | Rat 7 | Rat 8 | Rat 9 | |
| Osteopontin | OSTP_RAT | 6.00 | 0.038 | 0 | 0 | 4 | 7 | 9 | 8 | Yes |
| Calbindin | CALB1_RAT | 5.17 | 0.019 | 2 | 4 | 0 | 5 | 12 | 14 | Yes |
| Putative phospholipase B-like 2 | PLBL2_RAT | 3.50 | 0 | 1 | 2 | 1 | 5 | 5 | 4 | No |
| Cluster of Glyceraldehyde-3-phosphate dehydrogenase | G3P_RAT | 2.17 | 0.018 | 2 | 3 | 1 | 5 | 4 | 4 | Yes |
| CD166 antigen | CD166_RAT | 1.75 | 0.050 | 3 | 3 | 2 | 6 | 4 | 4 | Yes |
| Complement C3 | CO3_RAT | 1.71 | 0.045 | 9 | 8 | 7 | 10 | 17 | 14 | Yes |
| Beta-2-glycoprotein 1 | APOH_RAT | 1.70 | 0.048 | 3 | 4 | 3 | 5 | 5 | 7 | Yes |
| Na(+)/H(+) exchange regulatory cofactor NHE-RF3 | NHRF3_RAT | 1.58 | 0.026 | 7 | 6 | 11 | 12 | 14 | 12 | No |
| Copper transport protein ATOX1 | ATOX1_RAT | 1.53 | 0.013 | 6 | 4 | 5 | 7 | 7 | 9 | No |
| Regenerating islet-derived protein 3-gamma | REG3G_RAT | 0.46 | 0.003 | 18 | 16 | 14 | 8 | 8 | 6 | Yes |

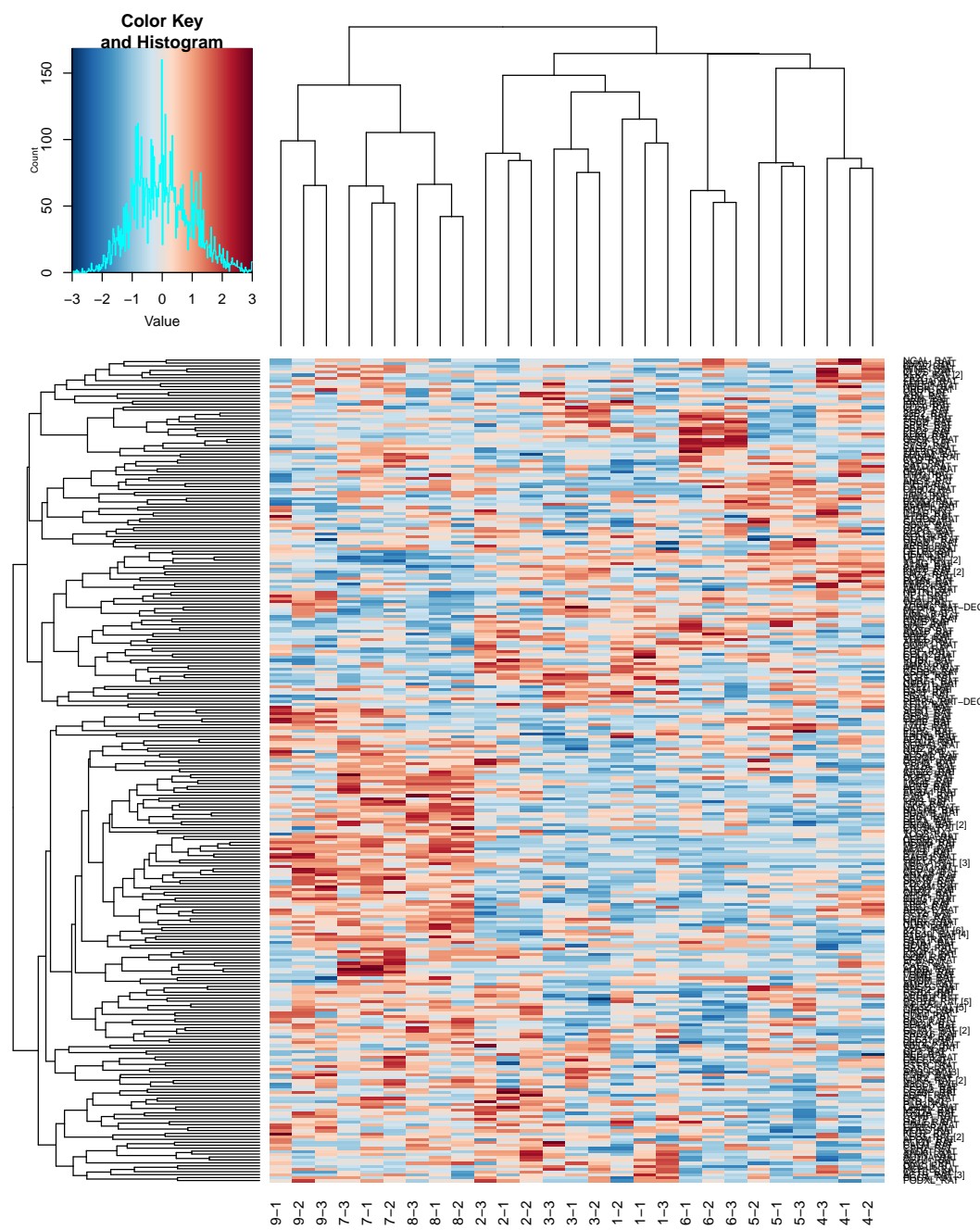

**Figure 3** **Heatmap of Hierarchical clustering.** 1, 2, 3, control group; 4, 5, 6, low dose group; 7, 8, 9, high dose group; −1, −2, −3, three technical replicates of each sample.

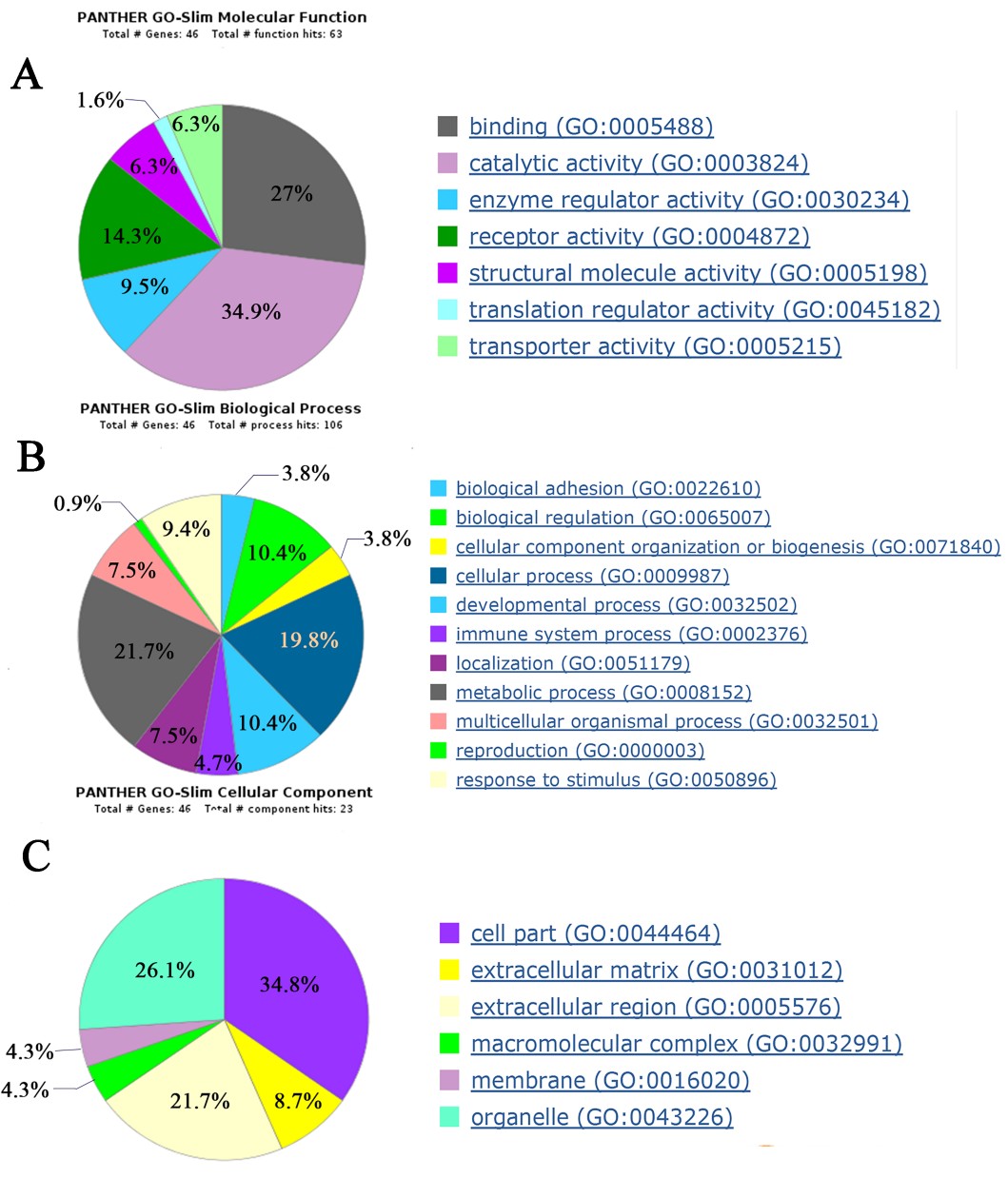

**Figure 4** **Biological function analysis of differential proteins between control group and AVP infusion groups.** (A) molecular functions of differential proteins; (B) biological processes of differential proteins; (C) cellular components of differential proteins.

It is well known that AVP can improve the permeability to water of the DCT and the connecting tubule and increase NaCl reabsorption. The $Na^{(+)}/H^{(+)}$ exchange regulatory cofactor NHE-RF3 plays an important role in regulating cell ion transport and membrane fluidity (*Kato et al., 2005*). NHE-RF3 also can mediate the formation of NHE3 (*Yang et al., 2014*), which is the major transporter in the proximal tubule that is involved in $Na^+$ reabsorption (*Schultheis et al., 1998*).

Calbindin in urine was 5.17-fold higher in the high-dose group than in the control group. NHE-RF3 in urine was 1.58-fold higher in the high-dose group than in the control group.

The changes of these three differential proteins, same as other differential proteins, were not significant after the $P$ values were adjusted by the Benjamini and Hochberger's method (*Benjamini & Hochberg, 1995*). The limitation of this result is that the candidate biomarkers should be further verified and filtered.

## Differential proteins as candidate biomarkers and their human orthologs

Nine of the 17 differential proteins have been identified as disease biomarkers in urine. Haptoglobin was identified as a biomarker for uranium nephrotoxicity (*Malard et al., 2009*), membranous nephropathy (*Ngai et al., 2007*), acute kidney injury (*Zager, Vijayan & Johnson, 2012*), diabetic nephropathy (*Rao et al., 2007*), bladder carcinoma (*Li et al., 2011*), hepatic fibrosis (*Van Swelm et al., 2013*) and acute phase response (*Piras et al., 2014*). Osteopontin can be used as a biomarker for ovarian cancer (*Rainczuk et al., 2013*), kidney stones (*Bautista et al., 1996*), bladder cancer (*Yang et al., 2011*) and drug-induced kidney injury (*Phillips et al., 2016*). Calbindin can serve as a biomarker for distal nephron segment injuries (*Iida et al., 2014*), nephrotoxicity (*Hoffmann et al., 2010*), acute kidney injury (*Togashi et al., 2012*) and drug-induced renal injury (*Fuchs et al., 2014*). Complement C3 can act as a biomarker of IgA nephropathy (*Liu et al., 2014*) and glomerulonephritis (*Cumming et al., 1976*). Pro-cathepsin H, regenerating islet-derived protein 3-gamma, glyceraldehyde-3-phosphate dehydrogenase, CD166 antigen and beta-2-glycoprotein 1 can be biomarkers of polycystic kidney disease (*Schaefer et al., 1996*), urinary tract infection (*Spencer et al., 2015*), ureteropelvic junction obstruction (*Mesrobian et al., 2010*), type 1 diabetes (*Suh et al., 2015*) and Dent's disease (*Cutillas et al., 2004*), respectively.

Sixteen of the 17 differential proteins have human orthologs. Twelve orthologs of these 16 proteins are part of the human core urinary proteome (Table 3). These results suggest that the effects of AVP on urinary proteins should be considered in future urinary disease biomarker researches.

## Candidate biomarkers for SIADH

SIADH is often confused with cerebral salt wasting syndrome (CSWS) in the clinical setting, leading to difficulty in confirming its diagnosis. Therefore, identifying effective biomarkers can facilitate the differential diagnosis of SIADH and then contribute to the effective treatment of hyponatremia that connected with increased mortality (*Corona et al., 2015*).The differential proteins between the control and AVP infusion groups of this study may be used as candidate biomarkers to aid in diagnosing SIADH. Large clinical samples are needed to determine whether these proteins can actually be utilized in clinical testing.

## DISCUSSION

Urine is a good source of biomarkers since it contains many changes excluded from the body. Urinary proteins are important bearer of information in urine. Many urinary protein biomarkers have been identified in various disease, such as renal disease (*Schanstra*

**Table 3  Human orthologs of the significant differential proteins between control group and AVP infusion groups.**

| Protein name | Rat protein accession number | Human ensembl gene ID | Human protein accession number | Human core urinary proteome |
|---|---|---|---|---|
| Haptoglobin | HPT_RAT | ENSG00000257017 | HPT_HUMAN | Yes |
| Prolactin-inducible protein homolog | PIP_RAT | ENSG00000159763 | PIP_HUMAN | Yes |
| Pro-cathepsin H | CATH_RAT | ENSG00000103811 | CATH_HUMAN | Yes |
| Junctional adhesion molecule A | JAM1_RAT | ENSG00000158769 | JAM1_HUMAN | Yes |
| Regenerating islet-derived protein 3-gamma | REG3G_RAT | ENSG00000172016 | REG3A_HUMAN | No |
| Glutamate–cysteine ligase catalytic subunit | GSH1_RAT | ENSG00000001084 | GSH1_HUMAN | No |
| Attractin | ATRN_RAT | ENSG00000088812 | ATRN_HUMAN | Yes |
| Malate dehydrogenase, cytoplasmic | MDHC_RAT | ENSG00000014641 | MDHC_HUMAN | Yes |
| Osteopontin | OSTP_RAT | ENSG00000118785 | OSTP_HUMAN | Yes |
| Calbindin | CALB1_RAT | ENSG00000104327 | CALB1_HUMAN | Yes |
| Putative phospholipase B-like 2 | PLBL2_RAT | ENSG00000151176 | PLBL2_HUMAN | Yes |
| Cluster of Glyceraldehyde-3-phosphate dehydrogenase | G3P_RAT | ENSG00000111640 | G3P_HUMAN | Yes |
| CD166 antigen | CD166_RAT | ENSG00000170017 | CD166_HUMAN | No |
| Beta-2-glycoprotein 1 | APOH_RAT | ENSG00000091583 | APOH_HUMAN | Yes |
| Na(+)/H(+) exchange regulatory cofactor NHE-RF3 | NHRF3_RAT | ENSG00000174827 | NHRF3_HUMAN | No |
| Copper transport protein ATOX1 | ATOX1_RAT | ENSG00000177556 | ATOX1_HUMAN | Yes |

& Mischak, 2015), bladder cancer (Grossman et al., 2005), preeclampsia (Carty et al., 2011) and cardiovascular diseases (Delles, Diez & Dominiczak, 2011). However some challenge exist in the urinary proteins biomarker studies such as standardization and normalization. The composition of the urinary proteins can be influenced by variation in sample handling process, including sample collection, protein extraction and protein digestion (Thongboonkerd, Chutipongtanate & Kanlaya, 2006). Standardized processes are considered to be beneficial for the integration analysis of the data from different sources. As for normalization, commercial software Scaffold (version 4.4.8; Proteome Software Inc., Portland, Oregon, USA) was used. The normalization method that Scaffold uses is to sum the "Unweighted Spectrum Counts" for each MS sample. These sums are then scaled so that they are all the same. The scaling factor for each sample is then applied to each protein group and adjusts its "Unweighted Spectrum Count" to a normalized "Quantitative Value".

Among the most significant differential proteins, only one protein was identified in both the low and high dose AVP groups, indicating that different mechanisms may be involved in responses to low and high doses of AVP, although both types of doses decrease urine volume and increase urine osmolality. In addition, we compared the results from this study with those of previous studies that investigated the effects of diuretics on urine (Li et al., 2014). Osteopontin excretion in urine significantly decreased after the rats were given the oral diuretic furosemide. In this study, osteopontin concentration was significantly elevated in urine from the high-dose AVP group, a result consistent with the finding that diuretics influence urine. Interestingly, haptoglobin concentrations in urine were significantly increased in both the high-dose AVP group in our study and in rats that were

given the oral diuretic furosemide. Thus, the anti-diuretic and diuretic share some of the same physiological processes and do not always exhibit opposing effects.

One challenge related to diagnosing SIADH is differentiating SIADH from CSWS because these two syndromes have similar clinical manifestations (hyponatraemia, high urine osmolality, and high natriuresis). It is important to distinguish SIADH from CSWS because these two syndromes differ with respect to pathogenesis and treatment (*Adrogue & Madias, 2012*). Measuring effective arterial blood volume is the main approach used to differentiate between these two diseases (*Palmer, 2000*). However, in clinical settings, it is difficult and expensive to determine effective arterial blood volume. Therefore, the efficiency of diagnosis and treatment would be improved by the identification of specific SIADH biomarkers. Differential proteins identified in this study can provide some clues for diagnose SIADH and future studies also be require to investigate biomarkers of CSWS and then aid the differential diagnosis.

The changes of the differential proteins (17 differential proteins) were not significant after the $P$ values were adjusted and the more stringent criteria ($P$ by ANOVA $\leq 0.05$, (2) fold change $\geq 1.5$, and (3) spectral count for each sample $\geq 4$) were used. However, the results before the $P$ correction can provide some meaningful clues for future researches that study the physiological mechanism of AVP and SIADH biomarkers.

Cancer is a common cause of SIADH. Various tumors, including lung cancer, pancreatic cancer, duodenal cancer, brain tumors and hematological malignancies, can cause SIADH (*Keenan, 1999*). Additionally, 7%–12% of small cell lung cancers are complicated by SIADH (*Berghmans, Paesmans & Body, 2000*). SIADH symptoms may occur before imaging existing tumors. Thus, SIADH may be used as an indicator for the early diagnosis of tumors, especially non-small cell lung cancer. Thus, the differential proteins identified in this study are helpful for the early detection of tumors and the diagnosis of SIADH after be validated in a larger blinded study with test specimens and validation specimens.

In this study, only the effect of AVP on male rats were analyzed. The previous study has demonstrated that vasopressin receptors gene AVPR2 is located on the X chromosome (*Juul et al., 2014*), suggesting that females may express more transcripts and receptors compared to males. Female rats should be included for future studies.

In conclusion, urinary proteins can be affected by AVP *in vivo*. Reports have indicated that several of the differential proteins identified in this study are associated with AVP; in addition, a number of the identified differential proteins have been recognized as disease biomarkers. These results suggest that several urinary biomarkers can be effected by AVP and thus we should consider the effects of AVP on urinary proteins in future urinary biomarker researches. The study data also provide clues regarding underlying mechanisms associated with AVP, and the identified differential proteins can be further studied to investigate their connections with AVP. Additionally, large clinical samples must be examined to verify that these differential proteins can actually be used as biomarkers of increases in pathological AVP in diseases such as SIADH.

### Funding

This work was supported by the National Key Research and Development Program of China (No. 2016 YFC 1306300), National Basic Research Program of China (973 Program) (No. 2013CB530805), Beijing Natural Science Foundation (Nos. 7173264, 7172076), the Fundamental Research Funds for the Central Universities (No.10300-31042110) and the Beijing Normal University (No. 11100704). The funders had no role in study design, data collection and analysis, decision to publish, or preparation of the manuscript.

### Grant Disclosures

The following grant information was disclosed by the authors:
National Key Research and Development Program of China: 2016 YFC 1306300.
National Basic Research Program of China (973 Program): 2013CB530805.
Beijing Natural Science Foundation: 7173264, 7172076.
Fundamental Research Funds for the Central Universities: 10300-31042110.
Beijing Normal University: 11100704.

### Competing Interests

The authors declare there are no competing interests. The work presented in this report is the subject of a pending patent.

### Author Contributions

- Manxia An performed the experiments, analyzed the data, contributed reagents/materials/analysis tools, wrote the paper, prepared figures and/or tables.
- Yanying Ni and Xundou Li contributed reagents/materials/analysis tools.
- Youhe Gao conceived and designed the experiments, reviewed drafts of the paper.

### Animal Ethics

The following information was supplied relating to ethical approvals (i.e., approving body and any reference numbers):

Institute of Basic Medical Sciences Animal Ethics Committee, Peking Union Medical College

Animal Welfare Assurance Number: ACUC-A02-2013-015.

### Patent Disclosures

The following patent dependencies were disclosed by the authors:

Pending patent. Name: Protein markers of syndrome of inappropriate antidiuretic hormone secretion;

Application number: 201610544477.5;

Date: 2016.07.12.

### Data Availability

An, Manxia; Ni, Yanying; Li, Xundou; Gao, Youhe (2017): Effects of arginine vasopressin on the urine proteome in rats. Figshare. https://doi.org/10.6084/m9.figshare.3622086.v1.

## Supplemental Information

Supplemental information for this article can be found online at http://dx.doi.org/10.7717/peerj.3350#supplemental-information.

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
