# Peer review of "Effects of arginine vasopressin on the urine proteome in rats"

_PeerJ, doi:10.7717/peerj.3350_

## Round 0.1 · original submission · Major Revisions

Please address all of the reviewers' concerns. I would also like you to provide more extensive description of your statistical procedures, specifically, whether or not your values include corrections for multiple-comparisons, as one would expect the levels of 5% of the proteins in every group (i.e. 21 in 420) to lie 2 standard deviations above the corresponding levels on the other group even if the "real", population-wide levels were equal on both groups.

Reviewer 1 ·

Basic reporting

The manuscript is very well organized and clear in most parts.

Experimental design

No comments

Validity of the findings

No Comments

Additional comments

Some sections and paragraphs need to be modified before it can be published. But it is just a matter of clarity.
Lines 53-67 I suggest to remove this introductive part about biomarkers.
Lines 97-115 Please state here how many animals were enrolled in the study, which ones provided the control samples, if the control samples were taken in different animals, in the same animal prior the AVP infusion. Please clarify as much as possible the whole experimental design.
Lines 128-145 Urine, as any biological specimen, is very sensitive to proteases and storage conditions. Please thoroughly describe the ways of sample collection, storage and preparation as in the following manuscript “Direct analytical sample quality assessment for biomarker investigation: Qualifying cerebrospinal fluid (DOI: 10.1002/pmic.201300565)”.
Lines 194-195 Again, experimental design is not clear, please rewrite this part making the description as clear as possible, from number of animals enrolled to sample collection.
Line 281: Authors interestingly describe the role of haptoglobin as biomarker in urine. However it is as well an acute phase protein and could be used as a biomarker of acute phase response (Serum protein profiling of early and advanced stage Crohn's disease. DOI: 10.1016/j.euprot.2014.02.010). I would suggest authors to consider this discussion in the light of the findings described in the indicated manuscript. Please take into account as well the fact that, if this protein is over-expressed in serum, could be released in higher amounts in urine and provide interesting information for other pathologies.

Reviewer 2 ·

Basic reporting

The manuscript is written in professional English with only some minor typographical errors.

Experimental design

A small number of animals (n=3 per group) were used for mass spectrometry. The animals were housed in metabolic cages which have been shown to cause stress in the animal, with subsequent hormonal modulation that could potentially bias the results. Collecting urine in a metabolic cage is convenient and more precise than other methods but the authors should discuss the potentially affects of the cages and housing conditions during this experiment.
A number of concerns with the experimental methods and results were noted which need to be addressed for robust and reproducible results. Additional experimental details are needed for the extraction of urinary proteins. Discrepancies in the text need to be addressed as specified in the Comments to the Author.

Validity of the findings

This rat model and label-free mass spectrometry experimental design is an early-stage, biomarker discovery platform. The proteins identified in this study are generally high abundance proteins, that have been associated with a variety of diseases/pathophysiologies. Thus any conclusions regarding their specificity for SIADH are highly speculative.

Additional comments

Using a rat model system, urine was analyzed by label-free mass spectrometry to provide insights regarding physiologic mechanisms associated with arginine vasopressin (AVP), an anti-diuretic hormone, and potential biomarkers for conditions with increased AVP production. The background regarding the pathophysiology of AVP and rationale for urine proteomic studies is appropriate in scope. Numerous papers have been published describing the syndrome of inappropriate antidiuretic hormone secretion (SIADH) in the clinical setting, however there are very few papers regarding biomarkers of SIADH.

Comments to authors
1. Line 54: The authors state that urine is not under homeostatic control. Renal function, and thus urine output, is indeed under homeostatic control. Renal filtration of the blood, reabsorption of solutes, and excretion of water are all examples of homoeostatic control of urine output. Urine output is also highly influenced by hydration status, exercise and diet. Further explanation of renal physiology and urine formation should be included.
2. Line 83: Please correct the statement regarding “increasing serum sodium levels” in patients with SIADH. SIADH results in hyponatremia (low serum sodium). This is stated correctly on line 287.
3. Line 90-91: Please explain the rationale for using a 1 week AVP infusion.
4. Metabolic cage housing could potentially bias the results of this study. Metabolic cage housing reduces a rodent’s ability to exercise, may lack nesting material, and grid flooring can cause injury to the hind feet. In addition, housing rats individually leads to social isolation which can elevate stress hormone levels (glucocorticoids and catecholamines), elevate blood pressure, heart rate and body temperature. See Kalliokoski O et al. Mice Do Not Habituate to Metabolism Cage Housing–A Three Week Study of Male BALB/c Mice PLoS One 2013 doi:10.1371/journal.pone.0058460.
5. Line 98: How many rats were in each group? Data for only 3 rats/group are shown in the Supplemental Table for the spectral counts, however in Figure 1, there are more than 3 rats per group. Please explain which rats were excluded/included in the mass spectrometry analysis and the rationale for not including all the rats.
6. Line 98: Only male rats were included in this study. Vasopressin receptors gene AVPR2 is located on the X chromosome, suggesting that females may express more transcripts and/or receptors compared to males (see Juul KV, et. al., American Journal of Physiology - Renal physiology. 2014 Vol. 306 no. 9, F931-F940 DOI: 10.1152/ajprenal.00604.2013.) Please provide the rationale for not including female rats and discuss this as a potential limitation of the study.
7. Lines 109-110: The physiological concentration of AVP in humans is 5-40pM. Please provide information regarding the AVP dosage for the rats and how it corresponds to the physiologic levels in normal and SIADH patients.
8. Lines 117-122: Many details are lacking regarding extraction of urinary proteins. This is a critical section of the manuscript because the urine is the biological fluid being analyzed. Pre-analytical and analytical variables could impact the final data. The urine was spun, debris removed, spun a second time to form a pellet and a supernatant. The pellet was lysed and the supernatant was used for acetone precipitation of soluble proteins. How was the acetone precipitate handled: was it concentrated, dried down, resolubilized in a buffer, etc.?
9. Lines 124-127: Coomassie Brilliant Blue may not be adequately sensitive to detect very low abundance proteins and/or differences between the control, high dose and low dose groups. A silver staining protocol should also be done to ensure that low abundance protein changes are noted. Was Coomassie Blue R250 or G250 dye used? The authors should see Winkler et al (Electrophoresis, 2007, PMID: 17516579 DOI:10.1002/elps.200600670) in which the authors compared Silver staining and Coomassie staining protocols for mass spec analysis and concluded that silver staining was more sensitive and better suited for nanoLC-MS.
10. Label-free spectral counting was used to measure protein abundance. However, the drawback of this method is that longer proteins may have more peptide identifications than shorter proteins. To compensate for this potential bias, a normalized spectral abundance approach should be used. See Zybailov B, et al. J. Proteome Res., 2006, 5 (9), pp 2339–2347. DOI: 10.1021/pr060161n.
11. Line 145: Each experiment was run in triplicate. Does this mean biological replicates, as in 3 different rats, or were these technical replicates, using 1 urine specimen that was divided into 3 aliquots?
12. Lines 175-177. What was the rationale for monitoring the rat’s body weight? The well-being of the rats may not be reflected by monitoring only body weight (Figure 1A). Changes in body temperature and fur state (smooth versus ruffled) can indicate stress/distress in rodents. (Kalliokoski O et al PLoS One 2013 doi:10.1371/journal.pone.0058460). The methods should include a description of how and when the body weights were measured.
13. Lines 199-208: Please provide more statistical details: which statistical software was used? How ere the 49 proteins determined to be changed between the control group and AVP infusion groups. What were the criteria for determining the change?
14. Lines 210-216 and Figure 3: The heatmap dendogram for the specimens indicate two main clusters: the high AVP dose cluster and a cluster containing both the control and low AVP dose groups. The statement that the three groups could be distinguished based on the heatmap” should be revised.
15. Line 244: Please explain where 17 differential proteins were identified. On line 200, 49 differential proteins were identified and there were 18 total differential proteins in the low and high AVP dose groups. Should 17 actually be 18?
16. Line 265: Please indicate the full name of CSWS at its first use.
17. Lines 278-284: The authors need to add a discussion of issues related to urinary protein biomarkers and normalization in urine specimens. The issue of normalization between urine specimens is particularly problematic for the field due to the high inter and intra-individual variation in urinary proteins. A simple urine protein:creatinine ratio be inadequate for normalization, whereas multiple analytes may be better to reduce variability between specimens. Also discuss how the urine biomarkers were normalized in the cited studies (Li et al 2014).
18. Line 301: The high abundance biomarkers identified in this study need to be validated in a larger blinded study with test specimens and validation specimens before they can be claimed as being “helpful for the early detection of tumors and the diagnosis of SIADH”. Tumor bearing rats were not studied and the effect of AVP on urine proteins was not studied in this disease cohort.
19. Figure 1. “Physiological indicators of rats in control group (n=7), low dose AVP group (n=5) and high dose AVP group (n=6).” Please explain why only 3 urine specimens from each group were analyzed (line 194) if there were additional animals in each group. More urine specimens would provide greater statistical significance. See also comment #6 above.

---

## Round 0.2 · Minor Revisions

Please address all reviewer requests.

Other review-level comments:

- line 108: Please change "The main process are descripted as follows. " to " The surgical procedure was performed as follows."

- The least significant difference (LSD) post-hoc analysis method used does not correct for multiple comparisons. Please include p-values corrected for multiple comparisons using either the method of Benjamini and Yekutieli (2001) ( http://www.jstor.org/stable/2674075 ) or family-wise false discovery rates (q-values) computed using Benjamini and Hochberger's method (http://www.stat.purdue.edu/~doerge/BIOINFORM.D/FALL06/Benjamini%20and%20Y%20FDR.pdf). Both these methods have been used by previous workers in the proteomics field to decrease false discovery rates (e.g. 10.1371/journal.pone.0139659 and 10.3390/proteomes1020040)

Reviewer 1 ·

Basic reporting

Authors significantly improved the overall quality of the manuscript which is now suitable for publication.
I suggest a final revision of the language and the correction of the following details:
Lines 83-84: Please add a reference here of AVP infusion as a rat SIADH model.
Line 135: “Thirty micrograms of protein from each sample was mixed with sample buffer and incubated at 96°C for 10 min.” Turn into: “Thirty micrograms of protein from each sample were mixed with sample buffer and incubated at 96°C for 10 min”. Same in line 141.
Line 211: Switch from: “Three rats each from the control, low-dose and high-dose groups were analyzed by LC-MS/MS” to “Urine specimens of three rats from the control, low-dose and high-dose groups were analyzed by LC-MS/MS”

Experimental design

No Comments

Validity of the findings

No Comments

Additional comments

No Comments

Reviewer 2 ·

Basic reporting

No comments

Experimental design

No comments

Validity of the findings

No comments

Additional comments

The authors addressed all the reviewer’s comments in the rebuttal letter. However they did not revise the manuscript for all the points. The comments listed below should be incorporated into the manuscript to improve clarity.
1. Regarding the homeostatic control of urine, the authors provided an explanation to the reviewer but did not revise the manuscript to provide clarity. The authors replied “We agree that urine output is also highly influenced by hydration status, exercise and diet. But when we stated that urine is not under homeostatic control, we meant that compare to urine, blood as the major component of internal environment, is under homeostatic control. Renal filtration, reabsorption of solutes, and excretion of water are actually all examples of homoeostatic control of internal environment. The changes we observed in urine are for maintaining the stability of the blood.” The important point that needs to be stated in the manuscript is: Renal filtration, reabsorption of solutes, and excretion of water are examples of homoeostatic control of blood which in turn affect the composition of urine.
2. Regarding the housing of the rats and urine collection in metabolic cages, the authors provided an explanation to the reviewer but did not revise the manuscript for clarity. The authors replied “Rats of all groups were housed in common feeding cages with every four to six rats in one cage. The rats were individually placed in metabolic cage only during the urine collection. We agree different conditions changed rats’ physiology. But all rats were in the same conditions and subjected to similar stress level during urine collection, even we are not sure if the same stress level would have exactly the same effect on rats of different groups. By the way, we did not observe any obvious injury to rats during urine collection.” Three important points that need to be added to the manuscript are: a) Rats of all groups were housed in common feeding cages with four to six rats in one cage. The rats were individually placed in a metabolic cage only during the urine collection; b) All rats were subjected to the same conditions and most likely experienced similar stress levels during urine collection; and c) No obvious injuries to the rats were observed during the urine collection in the metabolic cages.
3. Lines 207-209 SDS-PAGE analysis: The authors provided an explanation to the reviewer but did not revise the manuscript for clarity. The authors replied: “Urine proteins of the three rats in each group have been analyzed by SDS-PAGE before analyzed by LC-MS/MS. The SDS-PAGE gel show that there was no significant degradation of these samples (Figure 2) and then these samples be analyzed by LC-MS/MS.”, and also “In our study, 30microgram urinary proteins were analyzed in SDS-PAGE to observe the protein distribution. For each sample, 100microgram of the urinary proteins were digested directly in solution for LC-MS/MS analysis. We do not collect proteins from the gel based on the staining.” Please add new text to the SDS-PAGE Results section at lines 207-209 to cover these points: a) 30microgram of urinary proteins were analyzed in SDS-PAGE to observe the protein distribution; b) The SDS-PAGE gel showed that there was no significant degradation of these samples (Figure 2); and c) The gel bands were not excised for mass spectrometric analysis. A different aliquot (100 micrograms) of urinary proteins were analyzed by mass spectrometry.
4. Discussion, lines 298-300: The authors added information regarding normalization. “As for normalization, in this study, we actually compared the value of a specific protein/ total protein in disease group to its value in control group based on spectral count.” However this statement is unclear. Please state the actual mathematical equation(s) used for normalizing the spectral counts. What is meant by “value of a specific protein”? Is this the spectral count for an individual peptide divided by the total number of peptides identified in the “disease” group? What is the “disease group”? In this study, there were 3 groups of rats, controls, low AVP infusion, and high AVP infusion. The normalization information (equations) should be listed in the statistical section.

---

## Round 0.3 · Minor Revisions

Please address all remaining minor requests from the reviewer.

Reviewer 2 ·

Basic reporting

Lines 182-183: Please replace the parentheses with the correct heatmap version in R. “"...was performed and visualized by R’s gplot package to create heat maps via the heatmap.2() function”.

Line 196: Insert the word “was” after “P value”. the P value was equal to or less than 0.05.

Experimental design

no comment

Validity of the findings

Discussion: Text should be added to discuss the limitation of this study which is the loss of significance using the family-wise false discovery rates (q-values) computed using Benjamini and Hochberger's method.
Please add the following statements around line 338: The changes of the differential proteins (17 differential proteins) were not significant after the P values were adjusted and the more stringent criteria (P by ANOVA≤0.05, (2) fold change≥1.5, and (3) spectral count for each sample≥4)were used. However,the results before the P correction can provide some meaningful clues for future researches that study the physiological mechanism of AVP and SIADH biomarkers.

---

## Round 0.4 · Minor Revisions

I would have preferred to find a more extensive discussion of the limitations of your study (for example, by expanding on the tradeoff between type I and type II errors inherent in the multiple-adjustment corrections, or by explicitly stating the limitations in the abstract). Please include also the identity of the two proteins (mentioned in your rebuttal letter (Version 2)) which survive the Benjamini and Hochberger adjustment for multiple-comparisons.

---

## Round 0.5 · accepted · Accept

Requested corrections have been performed

Reviewer 2 ·

Basic reporting

No comment

Experimental design

No comment

Validity of the findings

No comment

Additional comments

The authors have made all the requested corrections.